# Doublecortin and Glypican-2 concentrations in the cerebrospinal fluid from infants are developmentally downregulated

**Catherine Brégère**[1⊙], **Urs Fisch**[1,2⊙], **Florian Samuel Halbeisen**[3], **Christian Schneider**[4], **Tanja Dittmar**[1], **Sarah Stricker**[5], **Soheila Aghlmandi**[3], **Raphael Guzman**[1,4,5,6]*

**1** Department of Biomedicine, Brain Ischemia and Regeneration, University Hospital Basel, Basel, Switzerland, **2** Department of Neurology, University Hospital Basel, Basel, Switzerland, **3** Basel Institute for Clinical Epidemiology and Biostatistics, University of Basel, Basel, Switzerland, **4** Division of Pediatric Neurosurgery, University of Basel Children's Hospital, Basel, Switzerland, **5** Department of Neurosurgery, University Hospital Basel, Basel, Switzerland, **6** Medical Faculty, University of Basel, Basel, Switzerland

⊙ These authors contributed equally to this work.
* Raphael.Guzman@usb.ch

**Data Availability Statement:** All relevant data are within the manuscript and its Supporting Information files.

## Abstract

### Objective

Doublecortin (DCX) and glypican-2 (GPC2) are neurodevelopmental proteins involved in the differentiation of neural stem/progenitor cells (NSPCs) to neurons, and are developmentally downregulated in neurons after birth. In this study, we investigated whether the concentrations of DCX and GPC2 in the cerebrospinal fluid (CSF) from human pediatric patients reflect this developmental process or are associated with cerebral damage or inflammatory markers.

### Methods

CSF was collected from pediatric patients requiring neurosurgical treatment. The concentrations of DCX, GPC2, neuron-specific enolase (NSE), S100 calcium-binding protein B (S100B), and cytokines (IL-1β, IL-2, IL-4, IL-6, IL-8, IL-10, IL-13, IFN-γ, and TNF-α) were measured using immunoassays.

### Results

From March 2013 until October 2018, 63 CSF samples were collected from 38 pediatric patients (20 females; 17 patients with repeated measurements); the median term born-adjusted age was 3.27 years [Q1: 0.31, Q3: 7.72]. The median concentration of DCX was 329 pg/ml [Q1: 192.5, Q3: 1179.6] and that of GPC2 was 26 pg/ml [Q1: 13.25, Q3: 149.25]. DCX and GPC2 concentrations independently significantly associated with age, and their concentration declined with advancing age, reaching undetectable levels at 0.3 years for DCX, and plateauing at 1.5 years for GPC2. Both DCX and GPC2 associated with hydrocephalus, NSE, IL-1β, IL-2, IL-8, IL-13. No relationship was found between sex, acute infection, S100B, IL-4, IL-6, IL-10, IFN-γ, TNF-α and DCX or GPC2, respectively.

**Funding:** This work was supported by the Department of Biomedicine, the Department of Surgery and the Innovation Focus on Pediatric Neurosurgery from the University Hospital Basel, Switzerland. Gottfried & Julia Bangerter-Rhyner Stiftung, Switzerland, also supported the study through a grant awarded to RG.

**Competing interests:** The funder had no role in the design of the study; in the collection, analyses, or interpretation of data; in the writing of the manuscript, or in the decision to publish the results. This study was performed and designed without the input or support of any pharmaceutical company or other commercial interest. The corresponding author had full access to all of the data in the study and takes responsibility for the integrity of the data and the accuracy of the data analysis. The authors report no disclosures.

## Conclusions

Concentrations of DCX and GPC2 in the CSF from pediatric patients are developmentally downregulated, with the highest concentrations measured at the earliest adjusted age, and reflect a neurodevelopmental stage rather than a particular disease state.

## Introduction

Neurogenesis refers to the generation of new neurons from neural stem/progenitor cells (NSPCs). It is a complex, timely, and spatially regulated process that involves multiple signaling pathways [1], and is nearly completed by birth, but continues in the early postnatal stages and during adult life in restricted areas of the brain, albeit in much lesser intensity. The disruption of neurogenesis due to genetic mutations or injuries during development or in the perinatal stage (hypoxic-ischemic injuries, inflammation) may contribute to severe neurological disorders [2]. Currently, the assessment of neurogenesis relies mainly on the analysis of postmortem brain tissue; despite attempts of monitoring in live beings through imaging [3, 4], reliable *in vivo* correlates of neurogenesis are still lacking, and thus represent a highly unmet medical need.

The cerebrospinal fluid (CSF) is a brain-specific fluid whose molecular composition may offer a window into ongoing physiological or pathological processes of the central nervous system (CNS). Doublecortin (DCX) and glypican-2 (GPC2), two neurodevelopmental proteins, have been detected and quantified in rodent and human CSF, and proposed as potential CSF biomarkers of neurogenesis [5, 6]. DCX is a microtubule-associated protein that plays an important role in the migration of newly generated neurons [7, 8]. Its expression in the brain marks the commitment of NSPCs into the neuronal lineage and lasts until the maturation of neurons. Factors that positively (e.g., exercise, environmental enrichment) or negatively (e.g., cranial radiation, aging) influence neurogenesis correspondingly enhance or downregulate the number of DCX expressing cells in the rodent adult hippocampal neurogenic niche [9–12]. As such, DCX is widely used as a marker of neurogenesis [13]. GPC2 is a glycosyl-phosphatidylinositol (GPI)-anchored heparan sulfate proteoglycan specifically expressed in the CNS during development; its expression is detected in young migrating postmitotic neurons [14, 15], and thus it marks a later stage of neurogenesis than DCX, as it is not present in mitotic neuroblasts. A recent study indicates that it may inhibit FGF-2-induced proliferation of NSPCs by binding this growth factor and that its expression in the brain is modulated by stimuli known to influence neurogenesis [5]. In the CSF from rat neonates and juveniles, the concentrations of DCX and GPC2 (hereafter referred to as CSF-DCX and CSF-GPC2, respectively) significantly decline during early postnatal stages, but while DCX reaches undetectable levels at around postnatal day 40 (P40) [16], GPC2 remains detectable and quantifiable at this time point [5]. There are a few reports of semi-quantitative DCX measurement in the CSF from human pediatric patients [17–21], and none of GPC2 available so far.

The specific association of these two proteins in the CSF with ongoing developmental, early postnatal, or adult- neurogenesis in the brain tissue remains nevertheless difficult to unambiguously establish, as CNS injuries/disorders or physiological development are accompanied by substantial cell death that may contribute to their presence in the CSF. Indeed, our own research in a rat model of neonatal hypoxia-ischemia (HI) indicated that CSF-DCX increased significantly 3 days after the brain HI injury, but this latter finding correlated with both HI-induced neurogenesis and infarct severity, thereby leaving unresolved whether a direct link exists between CSF-DCX and endogenous neurogenesis [6]. Accordingly, the goal of the

present study was to further investigate the possible association between the concentration of DCX and GPC2 in the CSF from pediatric patients and age as well as conventional markers of brain damage (NSE, S100B) and inflammation (cytokines).

## Methods

This observational study was performed at the University of Basel Children's Hospital and the University Hospital Basel, a Swiss tertiary academic medical care center. The STROBE guidelines (https://www.strobe-statement.org/) were followed to enhance the quality and standardization for the reporting of observational studies.

### Standard protocol approvals, registrations, and patient consents

The study was approved by the local ethics committee (Ethikkommission beider Basel EKBB, No. 42/10). Written informed consent was given by the legal custodian of all patients.

### Data and material collection

Patients less than 18 years old requiring a neurosurgical procedure involving the lumbar or cranial CSF compartment, performed at the Division of Pediatric Neurosurgery, University Children's Hospital of Basel, were eligible for this study. Patients were not included if the neurosurgical procedure by itself either did not involve the CSF compartment or if no CSF was collected during the neurosurgical procedure. CSF could be collected repeatedly per patient if they required multiple neurosurgical procedures, with each procedure requiring a new hospital admission. In one premature patient with extradural drainage due to ventricular hemorrhage, CSF was repeatedly collected from this neurosurgical procedure.

Lumbar or ventricular CSF was collected periprocedurally between 1st February 2013 and 30th November 2017, and kept at 4°C until further processing. CSF was then centrifuged at 4°C at 2000 g for 10 min, and the supernatant was aliquoted and stored at -80°C until analysis. CSF was classified as clear, xanthochrome (yellow) or bloody (reddish) after centrifugation by eye estimation.

Clinical data (patient sex, chronological age, premature birth status, main diagnosis, presence of symptomatic hydrocephalus, indication for neurosurgical intervention, type of intervention, and acute infection status) were extracted from individual patients' records at the time point corresponding to CSF collection. In prematurely born patients, adjusted age was calculated from chronological age and expected due date (gestational week 40 + 0). In this study, adjusted age is mentioned, unless stated otherwise. Main diagnoses, indications for neurosurgical intervention, and type of intervention are summarized in Table 1 and full details for all included patients are reported in S1 Table.

### Doublecortin and glypican-2 immunoassays

The two immunoassays were developed on the Meso Scale Discovery (MSD) platform. For the DCX assay, an MSD 96-well streptavidin-coated plate was blocked with 150 μl/well of blocking buffer (50 mM Tris/HCl pH 7.4, 60 mM NaCl, 0.1% Tween-20, 5% bovine serum albumin [BSA]) for 2 hours at room temperature without shaking. Plates were then coated overnight at 4°C with 25 μl of a biotinylated anti-DCX antibody (capture antibody, Abcam 77450, 0.4 μg/ml) in assay buffer (50 mM Tris pH 7.4, 60 mM NaCl, 0.5% BSA, and 0.1% Tween-20), and for two additional hours at room temperature on a rotational shaker (600 rpm) the next day. Plates were then washed three times with 150 μl/well of wash buffer (50 mM Tris pH 7.4, 60 mM NaCl, 0.1% Tween-20), and 50 μl of recombinant DCX (range between 0.61 to 10'000 pg/

**Table 1. Baseline characteristics of the 38 patients.**

| Characteristic | N | % |
|---|---|---|
| Female sex | 20 | 53% |
| Adjusted age at first measurement, yrs (median & IQR) | 3.28 | 0.31–7.7 |
| Preterm birth | 12 | 32% |
| Main diagnosis | | |
| Congenital aqueductal stenosis | 4 | 11% |
| Hydrocephalus malresorptivus | 7 | 18% |
| Spastic cerebral palsy | 7 | 18% |
| Symptomatic arachnoid cyst | 4 | 11% |
| CNS tumor | 5 | 13% |
| Other* | 11 | 29% |
| CSF origin | | |
| Arachnoid cyst | 2 | 5% |
| Spinal | 11 | 29% |
| Subdural | 1 | 3% |
| Ventricular | 24 | 63% |
| Hydrocephalus | 24 | 63% |
| Acute infection | 4 | 11% |

Characteristics are presented as absolute numbers and percentages, unless stated otherwise.

* The other diagnoses are described in S1 Table.

ml) or CSF samples (1:2 dilution) were incubated with the ruthenium-labelled m83 anti-DCX antibody (detection antibody, 0.16 μg/ml) on a rotational shaker (600 rpm) for 4 hours at room temperature. After disposal of samples and 3 washes, 150 μl/well of reading buffer (MSD read buffer T with surfactant) was added and the plate was immediately imaged on an MSD Sector Imager 6000. For the GPC2 assay, the same procedure was followed using biotinylated and ruthenium labelled anti-GPC2 antibodies (capture antibody: mAB2/32 1 μg/ml, Roche; detection antibody AF2304, R&D systems, 0.5 μg/ml) and recombinant GPC2 (R&D Systems 2304-GP, range between 4.11 to 3'000 pg/ml). The mean limit of detection (LOD) for each assay was 5.8 and 0.05 pg/ml for DCX and GPC2, respectively.

## Cytokine multiplex immunoassay

The Meso Scale Discovery assay V-Plex (Cat# K15210G-1) was used to measure levels of cytokines and chemokines, namely IFN-$\gamma$, IL-10, IL-13, IL-1$\beta$, IL-2, IL-4, IL-6, IL-8 and TNF-$\alpha$ according to the manufacturer's instructions. The corresponding LODs in pg/ml were 0.97, 0.13, 0.66, 0.04, 0.14, 0.06, 0.16, 0.1 and 0.26 respectively.

## NSE and S100B immunoassays

Human Neuron-specific Enolase (NSE) and S100B were measured using commercially available assays (Abcam ab217778 and Millipore EZHS100B-33K, respectively) and performed according to the manufacturer's instructions. The LODs in pg/ml were 97 and 2.7 respectively.

## Outcomes

The primary outcome was the association of DCX, or GPC2 in the CSF with patients' age. Secondary outcomes were the association of DCX with GPC2 and the association of DCX or

GPC2 with CSF markers for CNS damage or inflammation, or clinical features (sex, hydrocephalus, acute infection).

## Statistics

To summarize the baseline characteristics data of our study population, we used the total number and percentage of the total for categorical variables, and median and first- and third quartiles (i.e., interquartile range [IQR]) for continuous variables. In the main manuscript, the summary statistics provided for continuous variables are calculated using values at first measurement, to avoid misrepresenting the population by giving too much weight to the data from patients with repeated CSF sampling. It is noteworthy that for the assessment of the relationships between CSF-DCX, CSF-GPC2 and all other parameters, all values from each patient are taken into account, and not just values at first measurements.

In the analysis, we used the fully probabilistic (Bayesian) approach to use all the information of our small dataset containing censored data (i.e., data below the LOD) while taking into account the hierarchical nature of the data. The details of the models (e.g., model structure, assigned prior distributions) are provided in S1 File on statistical methods, and the most important information is briefly explained below.

To assess the relationship between DCX and GPC2 with age, we fitted an asymptotic mixed-effects regression model [22]. The asymptotic regression model is used to model the concentration of DCX and GPC2 approaching a long-term stable plateau (horizontal asymptote) after an initial decline. The model consists of three parameters, the intercept (the concentration at birth), the asymptote (concentration at the stable plateau) and the rate of change (the decline of concentration over time).

To assess the relationship between DCX and GPC2 with conventional markers of brain damage (NSE, S100B) and inflammation (IL-1β, IL-2, IL-4, IL-6, IL-8, IL-10, IL-13, IFN-γ, TNF-α), we fitted for each marker a univariable linear mixed-effects model with a random intercept to account for the multiple measurements per patient.

We fitted a univariable linear mixed-effects model with a random intercept to account for the repeated measurements per patient. In all models including measurements of DCX, we had measurements below the LOD of the assay. Using the fully probabilistic Bayesian approach, it is not necessary to ignore or to impute these censored values; instead, they can be incorporated into the model, where they are eventually integrated out of the posterior distribution [23]. For better visualization, in the graphs where DCX data are reported, the censored values are displayed using the mean value of the LOD of all DCX assays, i.e., 5.8 pg/ml.

Hemolysis is a very well-documented confounder of NSE determination [24]. Thus, in a sensitivity analysis, we excluded all samples with hemolysis visible to the naked eye from our analysis to assess the relationship between DCX and GPC2 with NSE.

All analyses were performed on the complete cases dataset, i.e., data entries with missing data (due to unavailability of CSF sample) were excluded. All the measured concentrations were log-transformed (natural logs). Data analysis was performed using R Version 3.6.1 (R Foundation for Statistical Computing, Vienna, Austria). To fit the models, we used the R package Bayesian Regression Models using Stan (brms) [25]. Additional details on the model specification can be found in S1 File on statistical methods.

## Results

### Baseline characteristics and CSF sample collection

A total of 63 CSF samples were collected from 38 patients. Among the patients, 21 had a single CSF sample collection, and 17 underwent repeated CSF sampling from multiple neurosurgical

procedures, totalling up to 41 CSF samples. Baseline characteristics of patients are shown in Table 1 with details of individual patient cases presented in S1 Table. While the youngest patients were mostly treated for intraventricular hemorrhage associated with prematurity around birth, older patients diagnosed with cerebral palsy underwent more often baclofen pump implantation.

## Determination of CSF-DCX, CSF-GPC2 and other analytes

DCX concentration was measured in the 63 CSF samples from all 38 patients, with a median concentration for the first available measurements of 329 pg/ml [Q1: 192.48, Q3: 1179.6]. CSF-DCX was below the LOD of the assay in 40 samples (63.5%) from 30 patients. The majority of samples below the LOD were from patients older than ~4 months. Only two patients older than 4 months had detectable DCX in the CSF: a 5-year-old male with a grade III astrocytoma (patient 140, CSF-DCX = 254 pg/ml) and a 9-year-old male with a teratoma of the III ventricle with obstructive hydrocephalus (patient 143, CSF-DCX = 329 pg/ml).

For the measurement of GPC2, all 59 available CSF samples from 38 patients were above the LOD. The median concentration of GPC2 for the first available measurements was 26 pg/ml [Q1: 13.25, Q3: 149.25].

All other analytes quantified are displayed together with DCX and GPC2 in Table 2. Their median concentration and interquartile range were also calculated using the first available measurements above LOD. The concentrations of all analytes for the 63 CSF samples are reported in S2 Table, and the summary statistics for the dataset that includes all measurements, i.e., all data above limit of detection per patient at different ages are shown in S3 Table.

## Association of CSF-DCX and CSF-GPC2 with age

After censoring the patients with CSF-DCX below the LOD, the association between log-transformed concentrations of DCX and age could be fitted in an asymptotic regression model, with the highest concentrations observed in the youngest patients, followed by a steep decline in DCX concentrations with advancing age (Fig 1 and S4 Table).

The relation between log-transformed concentrations of GPC2 and age could also be modelled with an asymptotic statistical fit (Fig 2 and S4 Table). The highest CSF-GPC2 values were measured in the preterm newborns. A decline in CSF-GPC2 was then observed in infants in

**Table 2. Data availability of variables and summary statistics at the 1st measurements.**

| Analyte | Total no. of CSF samples available | No. of measurements above detection limit | Median concentration (pg/ml) | Interquartile range (pg/ml) |
|---------|-----------------------------------:|-------------------------------------------:|-----------------------------:|-----------------------------|
| DCX | 63 | 11 | 329 | 192.48 to 1179.6 |
| GPC2 | 59 | 38 | 26 | 13.25 to 149.25 |
| NSE | 63 | 38 | 3461 | 1740.75 to 8626.5 |
| S100B | 54 | 35 | 1370 | 767.5 to 1852.5 |
| IL-1β | 59 | 36 | 0.21 | 0.09 to 0.56 |
| IL-2 | 59 | 33 | 0.51 | 0.27 to 2.27 |
| IL-4 | 59 | 22 | 0.13 | 0.08 to 0.52 |
| IL-6 | 59 | 38 | 2.78 | 1.24 to 8.12 |
| IL-8 | 59 | 38 | 56.29 | 22.98 to 81.84 |
| IL-10 | 59 | 35 | 0.31 | 0.19 to 0.56 |
| IL-13 | 59 | 30 | 1.69 | 0.78 to 3.95 |
| IFN-γ | 59 | 23 | 2.29 | 1.28 to 5.64 |
| TNF-α | 59 | 25 | 0.73 | 0.39 to 4.94 |

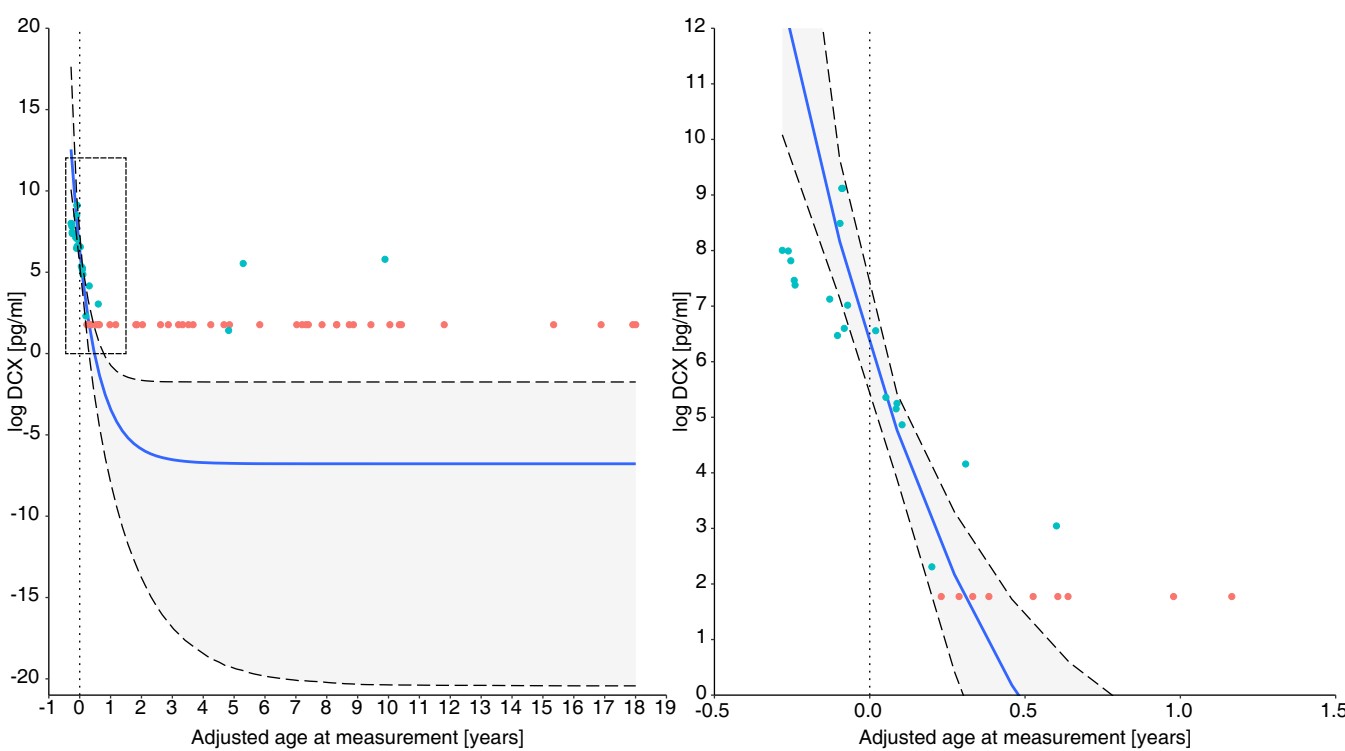

**Fig 1. Log concentration of DCX by term-born adjusted age at measurements (years).** The graph on the left shows the asymptotic relationship between CSF-DCX and age, and the right panel corresponds to the zoomed view of the dashed rectangle in the left graph. The blue line is the regression fitted curve and the grey area represents the 95% credible interval. Blue dots represent values over, and red dots below the limit of detection. The censored values are displayed as the mean value of the LOD of all DCX assays, and hence all align. One blue dot is below the mean LOD because it was within detectable range in the corresponding assay plate.

the age range of 0–1 year followed by a plateau in the age range of 1–18 years after 1.5 years with a median concentration value of 13.46 pg/ml.

## Longitudinal data on CSF-DCX and CSF-GPC2 in patients with repeated sampling

The relation between CSF-DCX/GPC2 and age was then examined in the 17 patients who underwent repeated CSF sampling. Overall, the age-related pattern in individual patients was consistent with that observed in the whole patient's cohort, namely a steep decline in CSF concentrations during prenatal/early postnatal stages followed by a stabilization of values, with either undetectable (for DCX) or very low levels (for GPC2) in older patients (Fig 3 and S1 Fig). Patient 141 though, for whom repeated sampling occurred within a very short time interval (3 days) had higher CSF-DCX/GPC2 at second and third sampling than at first collection time (S1 Fig).

## Association of CSF-DCX or CSF-GPC2 with other variables

There was a linear relationship between CSF-DCX and GPC2 (Fig 4 and S4 Table).

Significant, positive associations were found for both CSF-DCX and CSF-GPC2 respectively, with hydrocephalus, NSE, IL-2, IL-8, IL-13, and IL-1β. No associations were found for sex, acute infection, IL-4, IL-6, IL-10, IFN-γ, TNF-α, S100B, for both CSF-DCX and

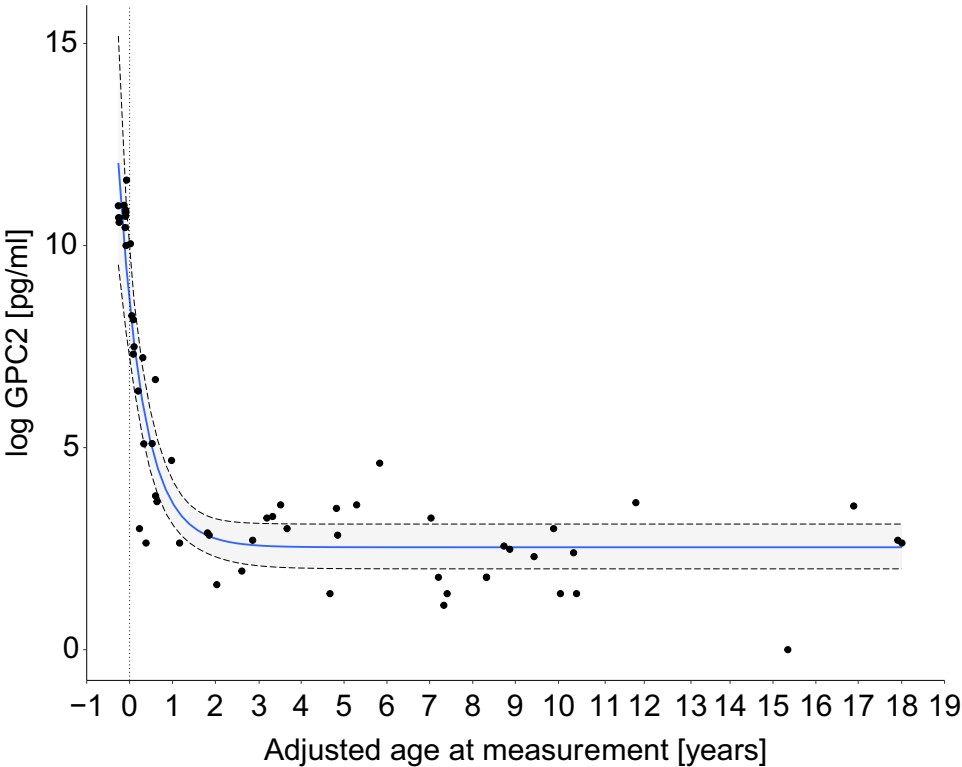

**Fig 2. Log concentration of GPC2 by term-born adjusted age at measurements (years) with an asymptotic regression model (blue line) and 95% credible interval (grey area).**

CSF-GPC2 (Table 3). The detailed parameters of the linear regression models assessing the relation between CSF-DCX/GPC2 and each clinical variable are provided in S5 Table.

Of note, the median age at CSF collection was significantly different among patients diagnosed or not with hydrocephalus, namely 0.455 years (~5 months) for CSF samples from hydrocephalus patients [Q1: -0.0793, Q3: 3.47] versus 8.32 years for CSF samples from patients with different diagnoses [Q1: 4.85, Q3: 10] (Fig 5).

We also further examined the association between CSF-DCX/GPC2 and NSE, in particular because hemolysis is a very well-documented confounder of NSE determination [24]. After running a sensitivity analysis excluding 16 non-clear CSF samples, the significant, positive association between NSE and either CSF-DCX or CSF-GPC2 still remained (S2 Fig and S2 File).

A question raised by the association between CSF-DCX/GPC2 and the CSF concentrations of NSE, IL-2, IL-8, IL-13, and IL-1β is whether the latter also decrease asymptotically with age. The relation between log-transformed CSF concentrations of IL-2 and age could be fitted in such asymptotic regression curve (S3 Fig). However, CSF-NSE, IL-8, IL-13 and IL-1β did not fit the pre-specified asymptotic decline. Graphs displaying log transformed concentrations of all measured analytes as a function of age are provided in S4 Fig.

## Discussion

The concentration of DCX and GPC2 in the CSF from pediatric patients correlated with age. The highest CSF concentrations of DCX and GPC2 were measured at birth and declined thereafter. CSF-DCX and CSF-GPC2 also correlated with hydrocephalus, the neuronal marker NSE, but not with the astrocyte marker S100B, acute infection or sex. CSF-DCX and

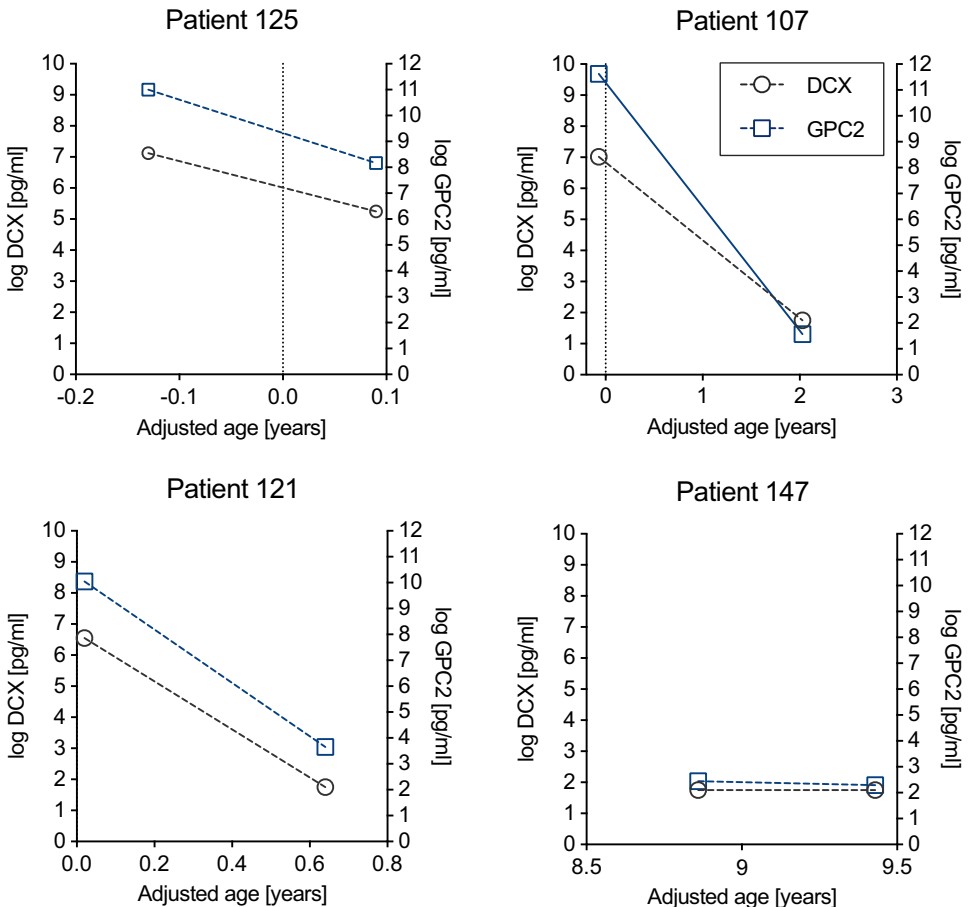

**Fig 3. The overall age-related decline in CSF-DCX and CSF-GPC2 can also be observed in individual patients with repeated CSF collection.** Graphs for patients 125, 107, 121 and 147 are displayed as representative examples. The DCX datapoints that were below the LOD are displayed as the mean LOD (5.8 pg/ml).

CSF-GPC2 also associated with individual cytokines, but not with others, independently of their classical pro- or anti-inflammatory categorization.

The age-related dynamics of CSF-DCX and CSF-GPC2 detected in the whole patient's cohort could also be retrieved in individual patients for whom CSF had been collected repeatedly. Patient 141 was nevertheless an exception, as increases in CSF-DCX and GPC2 were observed at second and third collection. This may be the consequence of CSF removal by serial tapping of ventricular reservoir within a very short time-lapse (three days), as described previously in the context of repeated CSF lumbar puncture [26], which may disturb CSF dynamics. While both DCX and GPC2 are two neurodevelopmental proteins, DCX is by far more characterized than GPC2 during neurodevelopment, in particular neurogenesis. The expression of DCX in the human brain tissue is high during CNS development, peaks around birth, and then dramatically declines in the early postnatal stages. In fact, "the relatively high expression in neonates compared to adults represents the largest and most significant change in gene expression found in the developing human brain with age" [27]. Thus, the age-related decline in CSF-DCX observed in our study agrees with that observation and lends support to our hypothesis that DCX in the CSF marks ongoing neurogenesis. CSF-DCX was almost always below the detection limit of our assay from around 4 months of age, an observation that parallels the finding that a population of DCX$^+$ young migrating neurons that gives rise to a

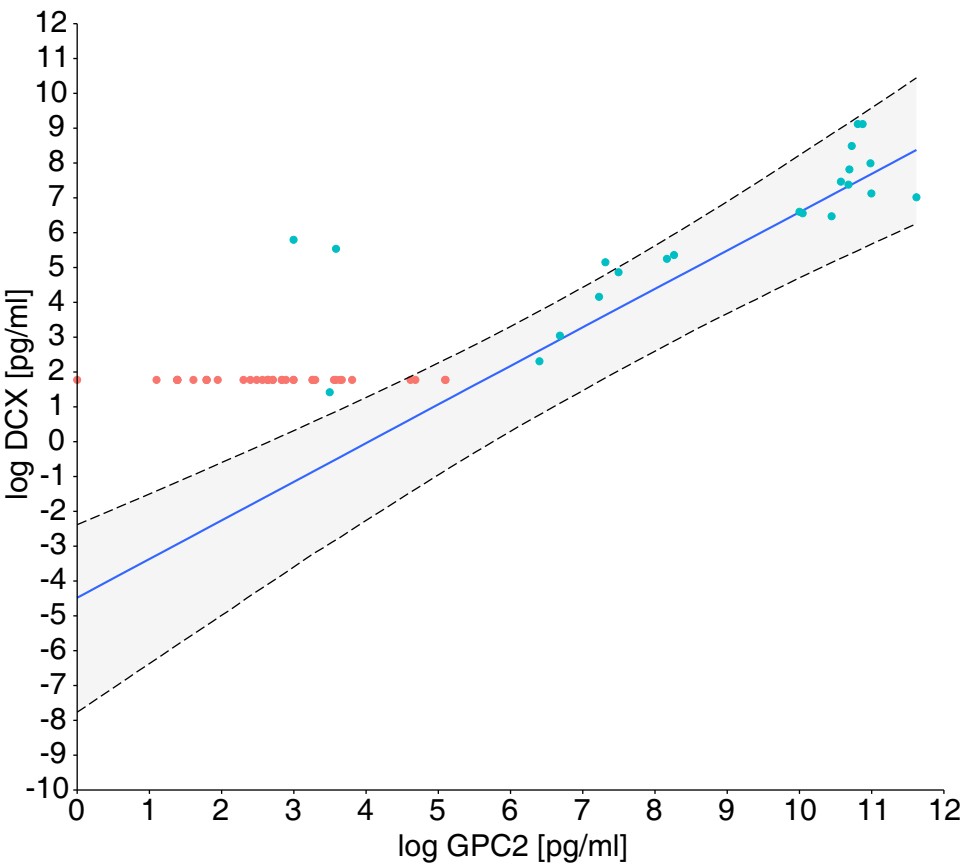

**Fig 4. Linear relation between log concentration of DCX and GPC2.** The graph shows the linear mixed-effect regression model (blue line) and 95% credible interval (grey area). Blue dots represent values over, and red dots below the LOD for DCX. As in Figs 1 and 3, DCX values below the LOD are displayed as the mean LOD of the DCX assays.

heterogeneous population of neurons in the anterior forebrain, persists in the human infant brain during at least the first 5 months of life [28]. Afterward, CSF-DCX remained mostly undetectable and thus, using our assay, DCX cannot be used to monitor post-neonatal neurogenesis. The two patients older than 4 months with measurable DCX levels were diagnosed with CNS tumors at the time of sample collection. DCX is known to be expressed in various CNS tumor tissue [29]. However, whether elevated CSF-DCX in these two patients was causally linked to the CNS tumor is beyond the scope of this investigation but merits further investigation. In the few reports that have examined the significance of DCX in the CSF from human pediatric patients, levels of DCX were suggested to reflect a neuroprotective response after traumatic brain injury, after nerve growth factor or methotrexate treatment [17–21]; Nevertheless, the number of patients was limited, and DCX was measured semi-quantitatively by western blot, without rigorous validation. Our data indicate rather that age is a stronger determinant of CSF-DCX than disease state.

Rodent studies show that the expression of GPC2 is high during development, peaks at birth, and then also declines in the brain to reach low levels during adulthood. Nevertheless, GPC2 can still be detected in the hippocampal dentate gyrus in the adult rodent [5]. While data on GPC2 expression in the human brain tissue is not available, our findings of an age-related decline in CSF-GPC2 are expected to reflect a corresponding age-related decline in the human brain tissue, similar to the rodent brain, although this warrants confirmation. In the

**Table 3. Relationship between Doublecortin and Glypican-2 and important clinical parameters-overview of univariable linear regression coefficients.**

| | Doublecortin | | | Glypican-2 | | |
|---|---|---|---|---|---|---|
| | **Estimate** | **Lower 95% CrI** | **Upper 95% CrI** | **Estimate** | **Lower 95% CrI** | **Upper 95% CrI** |
| Sex | -1.35 | -5.80 | 3.03 | -1.1 | -3.0 | 0.8 |
| Acute Infection present | 1.20 | -2.31 | 4.77 | -0.4 | -2.8 | 2.1 |
| Hydrocephalus present | **6.15** | **1.50** | **12.23** | **2.4** | **0.5** | **4.2** |
| NSE | **2.24** | **1.44** | **3.31** | **1.7** | **1.4** | **2.1** |
| S100B | 0.95 | -0.44 | 2.39 | 0.3 | -0.4 | 1.0 |
| IL-1β | **0.83** | **0.08** | **1.67** | **0.5** | **0.02** | **1.0** |
| IL-2 | **2.45** | **1.56** | **3.63** | **1.6** | **1.1** | **2.2** |
| IL-4 | -0.02 | -0.50 | 0.47 | -0.2 | -0.5 | 0.1 |
| IL-6 | 0.07 | -0.70 | 0.93 | 0.1 | -0.4 | 0.5 |
| IL-8 | **1.17** | **0.36** | **2.11** | **0.8** | **0.3** | **1.2** |
| IL-10 | 0.68 | -0.08 | 1.50 | 0.3 | -0.3 | 0.8 |
| IL-13 | **1.73** | **0.54** | **3.08** | **1.0** | **0.2** | **1.8** |
| IFN-γ | 0.81 | -0.22 | 1.96 | 0.5 | -0.3 | 1.4 |
| TNF-α | 0.42 | -0.07 | 0.99 | 0.2 | -0.2 | 0.6 |

Statistically significant associations are bolded.

CrI: Bayesian credible interval, there is a 95% probability that the true effect estimate is within the interval.

For all categorical variables (sex, acute infection, hydrocephalus), the estimate indicates the average difference in log analyte level between groups.

For the continuous variables (log-transformed NSE, S100B, cytokines), the estimate indicates how much increase in log DCX/GPC2 is associated with a given increase in the log analyte level. For instance, with an increase in log IL-1β concentration in CSF by 1 (equivalent of 2.718 pg/ml on the normal scale), the log concentration of DCX in the CSF increases by 0.83 log pg/ml (2.193 pg/ml on the normal scale).

report by Lugert et al., CSF-GPC2 was measured in the CSF from 83 human adults of different age groups (from the twenties to eighties) but did not show any particular correlation with age. Analysis of CSF samples from 10 individuals with repeated CSF sampling over 3 years follow-up still showed a slight age-related decline in 9/10 individuals, suggesting that GPC2 in the CSF may be a marker of adult neurogenesis.

CSF-DCX and GPC2 were associated with hydrocephalus, a condition characterized by an excess of CSF and causing an enlargement of the ventricles. It is most likely that the association between CSF-DCX/GPC2 and hydrocephalus is due to a confounding effect of age. Indeed, hydrocephalus was more common among younger patients in our cohort, thus limiting the reliability and clinical relevance of such association.

Our data revealed an association between CSF-DCX/GPC2 and NSE, but not S100B. Further analysis showed that CSF-NSE did not fit into a similar statistical model with age as DCX and GPC2. NSE is a cytoplasmic glycolytic enzyme highly expressed in neurons and neuroendocrine cells within the CNS [30]; during brain development in the rat, NSE expression increases continuously between birth and P40-P50 [31, 32], and coincides with functional neuronal maturation [30, 32]. S100B is a calcium-binding protein expressed in glial cells, notably astrocytes but also oligodendrocytes [33]. It exerts both intra- and extracellular functions and is implicated in diverse cellular activities, e.g., cell proliferation, differentiation, survival, signal transduction, but overall, its exact function during brain development remains undefined [34]. In the rat forebrain, S100B expression remains low between birth and P14 but then increases between P14 and P60, thus marking a mature developmental stage of the astrocytic lineage [35]. Both CSF-NSE and CSF-S100B are typically considered as markers of brain damage in diverse CNS conditions [33, 36, 37], and an elevation in both markers has been reported in infants suffering perinatal brain damage (for a review, see [38]). It may be argued the

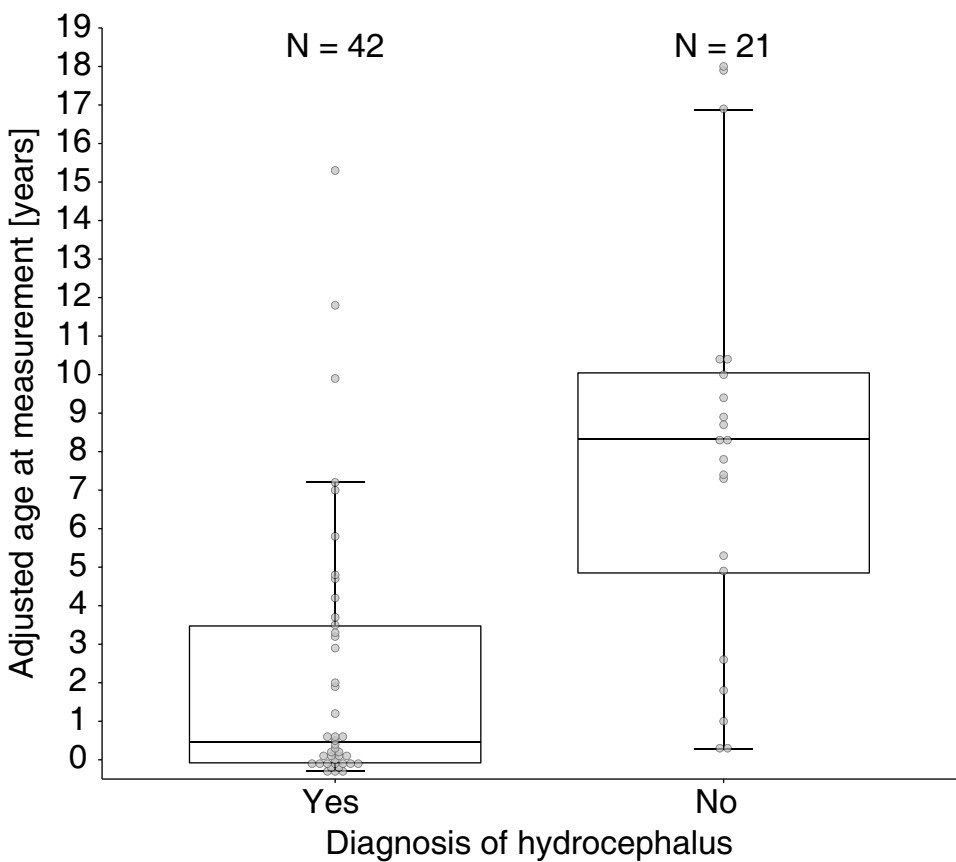

**Fig 5. Age at measurement (in years) among the 63 CSF samples from patients diagnosed with or without hydrocephalus.**

association between CSF-DCX/GPC2 and NSE is suggestive of DCX and GPC2 as markers of neuronal damage. However, in a sensitivity analysis, 16 CSF samples were excluded from the NSE data analysis due to hemolysis visible to the naked eye. Expectedly, the majority of these excluded samples (9/16), were from preterm newborns diagnosed with post-hemorrhagic hydrocephalus. The trustworthiness of the association between DCX and NSE is then particularly questionable since most CSF-DCX values are below detection limit after ~ 4 months of age, and 11 measurements are left to draw this association. Finally, free hemoglobin was not measured in our CSF samples, thus invisible hemolysis may still confound the NSE data to an unknown extent [24]. The absence of correlation between CSF-DCX/GPC2 and S100B, another specific marker of brain injury, further reduces the likelihood of a possible link to cerebral damage.

Overall, CSF-DCX/GPC2 were not associated with the clinical manifestations of infection of the patients, but an association was found with particular cytokines, namely IL-1β, IL-8, IL-2 and IL-13 for both DCX and GPC2. IL-1β, IL-8 and IL-2 are traditionally classified as pro-inflammatory cytokines, while IL-13 as anti-inflammatory, keeping in mind that some cytokines, depending on complex factors like environment or local concentration, display dual properties [39]. IL-1β is an important mediator of pro-inflammatory signaling, IL-8 is a chemokine involved in chemotaxis for neutrophils and in angiogenesis, IL-2 plays a role in T cell response, and IL-13 is considered as a T-cell helper type ($T_H$)-2 cytokine [40]. Correlation of CSF levels of cytokines are often made with particular disease parameters (e.g., severity of

disorder or neurological outcome [41–43]), but seldomly with non-cytokine/chemokine CSF proteins. In our study, the association between the aforementioned cytokines and DCX and GPC2 is difficult to interpret, especially in the absence of association with other pro- (IL-6, IL-10, IFN-γ, TNF-α) or anti-inflammatory (IL-4) cytokines. Notwithstanding, their median concentration values are within ranges of these reported by Pranzatelli et al. [44], with the chemokine IL-8 showing higher CSF concentration than cytokines. We agree with these authors that the low concentration of cytokines in CSF precludes reliable interpretation of data. Nevertheless, given the relationship between CSF-DCX/GPC2 and age, a compelling question was whether a similar relationship between the CSF concentrations of IL-1β, IL-8, IL-2 and IL-13 and age also existed. An asymptotic, age-associated decline in the CSF concentration of IL-2 was observed, but neither IL-1β, IL-8 nor IL-13 displayed such age-related pattern. Crucial roles of cytokines and their signaling pathways during CNS development are well-described [45]. IL-2 has been reported to impact morphology, survival, proliferation and differentiation of primary cultures of fetal neural cells [46]. But whether the decrease in CSF-IL-2 relates to specific neurodevelopmental events remains currently unclear.

The major strength of this study is the quantitative measurement of CSF-DCX and CSF-GPC2 in human pediatric patients, which, to the best of our knowledge, has not been performed previously. Limitations of this study include the small sample size, the heterogeneous clinical features of the patients, as well as the unavailability of CSF samples from healthy age-matched controls. Some baseline characteristics are nested such as CSF derived from a spinal tap collected mostly from older patients, or the prominent occurence of hydrocephalus in very young patients. Thus, there may be a selection bias that we cannot account for.

## Conclusion

The age-associated decline in DCX and GPC2 in the CSF parallels and reflects the well-documented developmental downregulation of these two proteins in the CNS tissue. While CSF-DCX has a limited temporal clinical utility for neurogenesis due to its undetectability with our immunoassay from around 4 months of age, its relevance could be explored in the context of CNS tumors. Unlike DCX, CSF-GPC2 can be detected at least until 18 years of age, making it a clinical candidate marker of neurogenesis. Since the exact mechanisms and circumstances of release of these two proteins into the CSF remain currently unknown, further investigations are warranted to evaluate the diagnostic and prognostic value of such markers in the human CSF.

## Supporting information

**S1 Fig. Longitudinal data on CSF-DCX and CSF-GPC2 in patients with repeated sampling.** (PDF)

**S2 Fig. Log concentration of NSE by term-born adjusted age at measurements (years).** (PDF)

**S3 Fig. Log concentration of IL-2 by term-born adjusted age at measurements (years), with an asymptotic regression model (blue line) and 95% credible interval (grey area).** (PDF)

**S4 Fig. Log concentrations of IL-8, IL-13, IL-1β, IL-4, IL-6, IL-10, IFN-γ, TNF-α and S100B by term-born adjusted age at measurements (years).** (PDF)

**S1 Table. Detailed clinical baseline characteristics of all 38 patients for each CSF sample.**
(PDF)

**S2 Table. Concentrations of DCX, GPC2, markers of neuronal damage and cytokines (in pg/ml) in each CSF sample from the 38 patients, along with corresponding age and main diagnosis.**
(PDF)

**S3 Table. Data availability of variables and summary statistics including all measurements from each patient.**
(PDF)

**S4 Table. Parameters of the asymptotic regression model assessing the relationship between DCX/GPC2 and age, and of the linear regression model assessing the relationship between DCX and GPC2.**
(PDF)

**S5 Table. Parameters of the linear regression models assessing the relation between CSF-DCX/GPC2 and each clinical variable.**
(PDF)

**S1 File. Statistical methods.**
(PDF)

**S2 File. The sensitivity analyses for the relations between CSF-DCX/GPC2 and NSE.**
(PDF)

## Acknowledgments

The authors thank Prof. Sven Wellmann MD (formerly University of Basel Children's Hospital, Basel, Switzerland; currently University Children's Hospital Regensburg, Germany) for providing CSF samples from a preterm newborn patient; Thomas Kremer Ph.D. and Sebastian Lugert Ph.D. (F. Hoffmann-La Roche AG, Pharma Research & Early Development, Basel, Switzerland) for providing antibodies against DCX and GPC2; Dr. rer. nat. Bernd Schwendele for assisting in figure preparation and Pia Bustos MSc for her excellent lab assistance.

## Author Contributions

**Conceptualization:** Catherine Brégère, Urs Fisch, Raphael Guzman.

**Data curation:** Catherine Brégère, Urs Fisch, Florian Samuel Halbeisen, Christian Schneider, Tanja Dittmar, Sarah Stricker, Soheila Aghlmandi.

**Formal analysis:** Catherine Brégère, Urs Fisch, Florian Samuel Halbeisen, Soheila Aghlmandi.

**Funding acquisition:** Raphael Guzman.

**Investigation:** Catherine Brégère, Urs Fisch, Tanja Dittmar.

**Methodology:** Catherine Brégère, Urs Fisch, Florian Samuel Halbeisen, Soheila Aghlmandi.

**Project administration:** Tanja Dittmar, Raphael Guzman.

**Supervision:** Catherine Brégère, Raphael Guzman.

**Validation:** Catherine Brégère, Soheila Aghlmandi.

**Visualization:** Catherine Brégère, Soheila Aghlmandi.

**Writing – original draft:** Catherine Brégère, Urs Fisch.

**Writing – review & editing:** Catherine Brégère, Urs Fisch, Florian Samuel Halbeisen, Soheila Aghlmandi, Raphael Guzman.

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
