## [Decision Letter · Decision Letter 0]

20 Jul 2022

PONE-D-22-09668Doublecortin and Glypican-2 concentrations in the cerebrospinal fluid from infants are developmentally downregulatedPLOS ONE

Dear Dr. Guzman,

Thank you for submitting your manuscript to PLOS ONE. After careful consideration, we feel that it has merit but does not fully meet PLOS ONE’s publication criteria as it currently stands. Therefore, we invite you to submit a revised version of the manuscript that addresses the points raised during the review process. Please respond to the points raised by the reviewer.

We look forward to receiving your revised manuscript.

Kind regards,

George Vousden

Deputy Editor in Chief

PLOS ONE

Journal Requirements:

Reviewers' comments:

Reviewer's Responses to Questions

**Comments to the Author**

1. Is the manuscript technically sound, and do the data support the conclusions?

Reviewer #1: Partly

2. Has the statistical analysis been performed appropriately and rigorously? 

Reviewer #1: Yes

3. Have the authors made all data underlying the findings in their manuscript fully available?

Reviewer #1: No

4. Is the manuscript presented in an intelligible fashion and written in standard English?

Reviewer #1: Yes

5. Review Comments to the Author

Reviewer #1: Neurogenesis plays a key role on brain development and homeostasis but there are not currently good indicators of the neurogenic state in a living animal. The cerebrospinal fluid, CSF, may reflect the physiological state of the central nervous system, as shown by the latest findings reporting cognitive benefits to adult mice after infusion of CSF from young mice.

Doublecortin (DCX) and Glypican-2 (GPC2) are key neurodevelopmental proteins involved in neurogenesis in mice, and have been proposed as potential reporters of neurogenesis. DCX and GPC2 have peak expression in the CSF perinatally and decline subsequently in early postnatal stages in rodents, but there are few or no reports on the CSF expression of those markers in humans.

Data for human patients is key to understanding human development and disease, but ethical experimental constraints limit their availability. This highlights the relevance of the present study by Guzman and colleagues, an approach to systematically investigate DCX and GPC2 expression in infant CSF. The same constraints affect this study, as data provided belongs to pediatric patients and is, therefore, partial and reflects infants requiring neurosurgery. But the information provided, together with future similar studies will help understand human neurogenesis.

The biggest strength of the study is the observation that human CSF expression of both factors reflects experimental data from rodents. The manuscript is well written and worthy of publication in PLOS ONE, with the authors nicely acknowledging the weak points of the study in the discussion. The reviewer would nevertheless suggest a few minor modifications to provide all available data to the public. In the reviewer’s view, this would improve the impact of the manuscript.

MajorPoints.

1. Full details of the clinical baseline characteristics for each patient are accounted for in the Supplementary Table 1, yet the information regarding the individual patient’s values for the DCX and GPC2, as well as the values for cytokines and markers for brain damage are not provided. In view of the reviewer, the information regarding the patient diagnosis accompanied by the individual measurements of DCX, GPC2 and the relationship with the individual values of the rest of molecules tested could provide a better understanding of their role.

2. Similarly, some of the patients were subject to repeated measures, but the respective measurements are not accounted for in the presented manuscript. In the reviewer’s opinion, this longitudinal information should be provided, as it would allow for better characterization of the dynamics of doublecortin and glypican-2 during human development, and in disease.

3. The reviewer wonders whether attributing the DCX values below the detection limits to the mean values of the lower detection limit to DCX concentration is generating an artifact to the data. Indeed, to the reviewer’s eye, the DCX data matches better a linear regression than an asymptote once the values that are below the detection limits of the assay (or data from patients with diseases of proliferative nature) are left out of the analysis. Would a randomized attribution of the values of DCX concentration –from 0 pg/ml to the lower detection limit– alter the interpretation of the data? Alternatively, a visual indication of the actual measurement should be included in the graph.

4. The researchers found positive correlation between the CSF levels of DCX and GPC-2 and the cytokines IL-1beta, IL-2, IL-8 and IL-13, and the marker for brain damage NSE. The authors provide a graph showing that log-transformed measurements of NSE do not show an asymptotic decline with age. The authors state that a visual assessment of the relation between log-transformed concentrations of cytokines with age indicated an absence of relation between those parameters. The authors provide the graphs for IL-1�, IL-2, IL-8 and IL-13. To the reviewer’s eye, these show that maximum expression levels appear perinatally to later stabilize. On the other hand, the authors do not display the same data for the remaining interleukins and the marker S100B. In view of the reviewer, these graphs should also be provided.

Minor Points.

1. A below-detection limit value is present in the figure-1 graph.

2. S2 table is labelled as “S1”. S4 tables are labelled as “S3”.

6. PLOS authors have the option to publish the peer review history of their article (what does this mean?). If published, this will include your full peer review and any attached files.

Reviewer #1: No

---

## [Author Response · Author response to Decision Letter 0]

7 Sep 2022

Thank you for allowing us to submit a revised version of our manuscript. 

The valuable insights of the reviewer have significantly improved our paper.

Our point-by-point response to the reviewer is presented in the file “Response to Reviewers”.

---

## [Editor Report · Decision Letter 1]

6 Dec 2022

Doublecortin and Glypican-2 concentrations in the cerebrospinal fluid from infants are developmentally downregulated

PONE-D-22-09668R1

Dear Dr. Guzman,

We’re pleased to inform you that your manuscript has been judged scientifically suitable for publication and will be formally accepted for publication once it meets all outstanding technical requirements.

Kind regards,

Robert Blum

Academic Editor

PLOS ONE

Additional Editor Comments (optional):

Dear authors,

I took over the mansucript as academic editor. There was some delay due to a lack of response by the reviewer. Anyhow, the reviewer gave great comments.

Indeed, this paper is important and provides a clear message. Data handling is transparent and you answered all reviewer comments.

For these reasons, I decided to terminate the review process and decided to recommend your work for publication in PLOSone.

Kind regards

Robert Blum (academic editor)
---

## [Editor Report · Acceptance letter]

5 Jan 2023

PONE-D-22-09668R1 

Doublecortin and Glypican-2 concentrations in the cerebrospinal fluid from infants are developmentally downregulated 

Dear Dr. Guzman:

I'm pleased to inform you that your manuscript has been deemed suitable for publication in PLOS ONE. Congratulations! Your manuscript is now with our production department. 

Kind regards, 

on behalf of

PD Dr. Robert Blum 

Academic Editor

PLOS ONE